

# A global daily mesoscale front dataset from satellite observations: In situ validation and cross-dataset comparison

Qinwang Xing[1], Haiqing Yu[2], Wei Yu[1,3,4,5], Xinjun Chen[1,3,4,5], Hui Wang[2,6,7]

[1]College of Marine Living Resource Sciences and Management, Shanghai Ocean University, Shanghai, 201306, China
[2]Institute of Marine Science and Technology, Shandong University, Qingdao, 266237, China
[3]National Engineering Research Center for Oceanic Fisheries, Shanghai Ocean University, Shanghai, 201306, China
[4]Key Laboratory of Sustainable Exploitation of Oceanic Fisheries Resources, Ministry of Education, Shanghai Ocean University, Shanghai, 201306, China
[5]Key Laboratory of Oceanic Fisheries Exploration, Ministry of Agriculture and Rural Affairs, Shanghai, 201306, China
[6]Southern Marine Science and Engineering Guangdong Laboratory (Zhuhai), Zhuhai, 519080, China
[7]National Marine Environmental Forecasting Center, Beijing, 100086, China

*Correspondence to*: Haiqing Yu (yuhaiqing@sdu.edu.cn), Wei Yu (wyu@shou.edu.cn), Xinjun Chen (xjchen@shou.edu.cn)

**Abstract.** Ocean fronts have garnered significant attention from researchers across various scientific disciplines due to their profound ecological and climatic impacts. The development of front detection algorithms has enabled the automatic extraction of frontal information from satellite observations, providing valuable tools for understanding the biophysical interactions within marine ecosystems. However, the lack of comprehensive validation and comparison of cross-satellite products against in-situ observations, along with limited accessibility to frontal datasets, must be addressed to enable the broader application of front detection algorithms. This study promoted the improved histogram-based front detection algorithm to global oceans with additional enhancements, generating the first publicly available, high-resolution, daily global mesoscale front dataset spanning from 1982 to 2023 (Xing et al., 2024a, https://doi.org/10.5281/zenodo.14373832). Global validation using in-situ underway observations shows that most in-situ and satellite-detected fronts can be matched with each other, with high temporal and spatial consistency, demonstrating the dataset's acceptable performance in detecting fronts. Cross-dataset comparisons reveal that multi-satellite merged products offer the best front detection performance, followed by observation-assimilated ocean model products, while single-satellite and purely simulated products show the lowest performance, all of which provide independent validation of the satellite-based global occurrence patterns. These results enhance confidence in the application of satellite-based front detection, and our global front dataset and detection algorithm may be valuable for both regional and global studies in marine ecology, fisheries, ocean dynamics, and climate change.

## 1 Introduction

Ocean fronts are typically narrow, high-gradient three-dimensional transition zones in the ocean where water properties (such as temperature, salinity, density, and chlorophyll) exhibit sharp variations over short distances (Belkin et al., 2009; Fedorov, 1986). These fronts are ubiquitous, high-energy dynamic processes in the global upper ocean that can alter physical, ecological, and biogeochemical processes of marine ecosystems through strong water mass convergence, secondary circulation, turbulent mixing, and other dynamic mechanisms (Mcwilliams, 2021). The significant characteristics of ocean fronts are their strong vertical mixing and retention capacity, which brings nutrients into the euphotic layer, fuelling phytoplankton bloom and further creating biological hotspots (Belkin et al., 2009; Ito et al., 2023). The accumulation of floating organisms around ocean fronts attracts and aggregates higher trophic levels, establishing productive food webs in these regions (Belkin, 2021; Woodson and Litvin, 2015). These processes significantly regulate air-ocean interactions and affect global climate change by enhancing the exchange of heat, oxygen, carbon, and other climatically important gases (D'Asaro et al., 2011). Additionally, recent research has found that frontal occurrences can be influenced by the El Niño-Southern Oscillation in certain upwelling systems (Amos and Castelao, 2022; Zhang et al., 2023), and frontal occurrences may further strengthen with global warming in some boundary current systems and upwelling systems (Xing et al., 2024b). Over the past few decades, significant attention has been devoted



to studying ocean fronts oceanographers, climatologists, ecologists, and fishery scientists, seeking to elucidate front-driven biophysical coupling mechanisms and their effects on climate change.

In earlier studies, research on front dynamic processes and their ecological and climatic effects primarily focused on specific
cases of known fronts through field surveys and numerical modelling (Alemany et al., 2009; Jing et al., 2015; Munk et al., 2010; Stukel et al., 2017; Taylor and Ferrari, 2011). The advancement of satellite Earth observation, combined with automated detection algorithms, has introduced new technology for elucidating front-driven changes in marine ecosystems (Belkin and O'Reilly, 2009; Castelao et al., 2006; Cayula and Cornillon, 1992). These technologies are capable of simultaneously observing most or even all of the horizontal structure of fronts in the global ocean and recording their historical variations over recent
decades (Mauzole, 2022; Xing et al., 2023; Xing et al., 2024b), providing a cost-effective alternative to high-cost in-situ measurements and labour-intensive manual detection of satellite images. Information on front occurrence derived from satellite-based automatic detection algorithms can be easily matched with other physical, ecological, and biological data in the areas of interest, allowing for broader applications beyond case studies of specific fronts. Recently, front detection algorithms have recently found increasing application across diverse fields, including marine ecology (Cox et al., 2016; Queiroz et al.,
2016), marine protection (Miller et al., 2015; Scales et al., 2014), fisheries (Woodson et al., 2012; Xu et al., 2017), marine pollution (Le et al., 2024), climate change (Kahru et al., 2018; Xing et al., 2024b), ocean dynamics (Amos and Castelao, 2022; Belkin et al., 2024; Wang et al., 2021).

As studies on ocean fronts continue to grow, the accessibility of front data has become a significant barrier for further investigations. The National Oceanic and Atmospheric Administration (NOAA) has provided a near real-time dataset of frontal
gradient magnitude and direction since 2012, which specifically covers the coastal areas of the United States. Sudre et al. (2023) developed a high-resolution thermal front dataset for the Mediterranean Sea and the southwest Indian Ocean, spanning from 2003 to 2020. However, while these datasets are derived using the Belkin-O'Reilly Algorithm and cover limited regions and time periods (Belkin and O'Reilly, 2009), to our knowledge, there is currently no publicly available global dataset of ocean fronts spanning the past few decades that utilizes the histogram-based Cayula and Cornillon algorithm (CCA)—the most
widely used detection algorithm in marine ecology and fisheries (Cayula and Cornillon, 1992; Cayula and Cornillon, 1995). Researchers from disciplines outside satellite observations often encounter a steep learning curve, requiring considerable time and effort to obtain the front information they need, which poses a significant challenge for many ecologists and fisheries scientists. In particular, the histogram-based CCA and its subsequent improvements involve complex calculations, demanding a high level of proficiency and substantial computational resources. Although the original CCA has been integrated into the
Marine Geospatial Ecology Tools to assist researchers in obtaining front information from satellite observations (Roberts et al., 2010), it does not include the more recent improved algorithms (Nieto et al., 2012), particularly the one proposed by Xing et al. (2023a), which effectively addresses some known limitations of the original CCA, such as undetected coastal fronts, repeated detections, and discontinuities. In comparison, research on ocean eddies, another key dynamic process, has experienced explosive growth over the last decade following the release of the first public global eddy dataset (Chelton et al.,
2011). Therefore, it is imperative to create an open-access global front dataset to address the increasing demands of front-related interdisciplinary research and applications.

The statistical analysis that integrates frontal indicators with other environmental and biological data relies heavily on reliable information regarding front occurrences; thus, the robustness of this data is fundamental for its application in ecology and fisheries research. Front occurrence data derived from satellite-based automatic detection can be affected by multiple factors,
including the limitations of the detection algorithms and errors in satellite-observed data. Therefore, such data needs to be further validated by ship-based in-situ observations. Ullman and Cornillon (2000) and Chang and Cornillon (2015) used continuous temperature measurements from a container ship regularly navigating between Port Elizabeth, NJ and Bermuda to validate CCA in the North Atlantic Ocean, while Miller (2009) used manually annotated front locations in the coastal waters of the northwest Iberian Peninsula to validate satellite-based frontal detection. While front detection algorithms, such as CCA,



have been widely applied to global waters, their reliability has not been comprehensively validated using in-situ data across the broader global ocean. Aside from a few well-studied regions, this leaves uncertainty about whether they perform consistently well in other areas. Additionally, previous studies have applied CCA to various SST datasets, including MODIS Level 3 SST, AVHRR Level 3 SST, and multi-satellite reanalysed Level 4 SST (Mauzole, 2022; Xing et al., 2023b). Frontal detection can be significantly affected by cloud contamination in Level 3 SST data, potentially leading to misinterpretation of frontal variations (Suberg et al., 2019). Previous studies have proposed different global frontal occurrence patterns based on varying datasets (Mauzole, 2022; Xing et al., 2023b), and the reasons for these differences remain a subject of debate. However, few studies have systematically compared detection performance across different SST datasets. Such comparisons are essential to guide researchers in selecting the most suitable SST data for obtaining reliable front information.

In this study, we applied additional modifications and enhancements to the recently improved CCA proposed by Xing et al. (2023a), extending it to global satellite-observed sea surface temperature (SST) to create a publicly accessible, high-resolution, daily ocean front dataset spanning the past 42 years (Xing et al., 2024a, https://doi.org/10.5281/zenodo.14373832). This dataset is readily downloadable and intended for broad applications in ecology, fisheries, and oceanography research. To comprehensively validate and assess the spatiotemporal reliability of both the dataset and detection algorithm, we used an extensive set of global in-situ sea surface underway measurements collected over the past 35 years. These in-situ data further allowed us to evaluate the performance of frontal detection across various independent SST products, including multiple satellite, ocean model, and reanalysis SST datasets (see Table 1), supporting the robustness of our global front detection algorithm and data product.

## 2 Data

Five independent SST datasets were used for front detection in this study to conduct cross-dataset validation with underway observations, all of which have been widely applied in previous oceanographic and ecological research. Each dataset offers its own advantages and represents different types (Table 1). Based on the validation results with global underway data, we selected the optimal SST dataset to determine the final global front dataset.

Table 1: Information on the different SST datasets used for front detection in this study.

| SST product | Type | Time range | Spatial resolution | Temporal resolution | Website |
|---|---|---|---|---|---|
| ESA CCI | Multi-satellite analysis | 1982-2023 | ~5 km | 1 d | https://dx.doi.org/10.5285/4a9654136 a7148e39b7feb56f8bb02d2 |
| MODIS | Cloud-contaminated single-satellite observation | 2003-2023 | ~4 km | 8 d | https://doi.org/10.5067/MODAM-8D4N9 |
| REMSS | Microwave and infrared analysis | 2003-2023 | ~9 km | 1 d | https://data.remss.com/SST/daily/mw _ir/v05.1/ |
| GLORYS | Observation-assimilated model | 1993-2023 | ~8 km | 1 d | https://doi.org/10.48670/moi-00021 |
| HadGEM3 | Pure ocean simulation | 1982-2023 | ~8 km | 1 d | https://esgf-node.llnl.gov/projects/cmip6/ |

### 2.1 Satellite-observed dataset

Our global mesoscale front dataset is constructed from a multi-satellite reprocessed Level 4 SST dataset derived from the version 3 Climate Data Record produced by European Space Agency's Climate Change Initiative SST project (ESA SST CCI CDRv3, https://dx.doi.org/10.5285/4a9654136a7148e39b7feb56f8bb02d2). This dataset provides 0.05° × 0.05° cloud-free



daily mean SST and sea ice concentration data spanning from 1982 to 2023. It is generated by the Operational Sea Surface
       Temperature and Ice Analysis (OSTIA) system, which merges three series of thermal infrared sensors (AVHRRs, ATSRs, and
       SLSTRs) with two microwave sensors (AMSR). This dataset has undergone extensive long-term validation against a large
       number of global independent in-site observations and is deemed suitable for climate applications (Embury et al., 2024).

       The MODIS Aqua night-time L3 SST dataset from NASA was also utilized to detect global fronts, serving as a single-satellite
cloud-contaminated product for dataset comparison (https://doi.org/10.5067/MODAM-8D4N9). This dataset is derived from
       the mid-infrared wavelength channels, with a spatial resolution of 0.0417° × 0.0417° and a temporal resolution of 8 days,
       covering the period from 2003 to 2023 (Kilpatrick et al., 2015). Furthermore, a satellite-merged Level 4 SST analysis, version
       5.1, from Remote Sensing Systems (REMSS) was adopted for global front detection as a comparative dataset representing an
       analysis product (https://doi.org/10.5067/GHMWI-4FR05). This dataset is generated from five through-cloud microwave
sensors (TMI, AMSR-E, AMSR2, WindSat, GMI) and four high-resolution infrared sensors (MODIS-Terra, MODIS-Aqua,
       VIIRS-NPP, VIIRS-N20) through optimal interpolation, providing 0.0879° × 0.0879° cloud-free daily SST spanning 2003 to
       2023.

### 2.2 Reanalysis and numerical simulation dataset

       The daily reanalysis SST dataset was extracted from Global Ocean Physics Reanalysis (GLORYS12) provided by CMEMS to
detect and validate global fronts, serving as a comparison with observation-assimilated models. This dataset is derived from
       the eddy-resolving (1/12°) NEMO platform, covering the period from 1993 to 2023, and is forced by ECMWF ERA-Interim
       and ERA5 reanalysis products. It also assimilates various ocean parameters obtained from in-situ observations and satellite
       datasets by a reduced-order Kalman filter (Jean-Michel et al., 2021). The daily eddy-resolving historical simulations from the
       CMIP6 HighResMIP experiment were also employed for cross-dataset front validations, serving as a pure model comparison
(Roberts et al., 2019). This simulation is based on the HadGEM3-GC31-HH model from the Met Office Hadley Centre, with
       a nominal 8 km resolution, covering the period from 1982 to 2023.

### 2.3 Underway data

       To compare and validate satellite-based front detection, we used global in-situ SST data from the Surface Underway Marine
       Database (SUMD), provided by NOAA NCEI (https://www.ncei.noaa.gov/archive/accession/NCEI-SUMD). This dataset is
sourced from thermosalinographs, meteorological packages, and other sensors deployed on over 450 ships and unmanned
       surface vehicles. All SST data in each sensor's trajectory have undergone standardized quality control procedures and criteria.
       Figure 1 shows the spatial distribution of the SUMD SST data utilized in this study, covering nearly all global regions. The
       highest density of SST records is found in the North Atlantic Ocean, with lower densities in the Southern Ocean, Arctic Ocean,
       and Indian Ocean. Most SST records are concentrated between 10°N and 40°N. The SST data from SUMD have a high
observation frequency, varying from seconds to minutes, resulting in spatial resolutions typically less than 1 km, even though
       the sailing speeds may differ among trajectories.

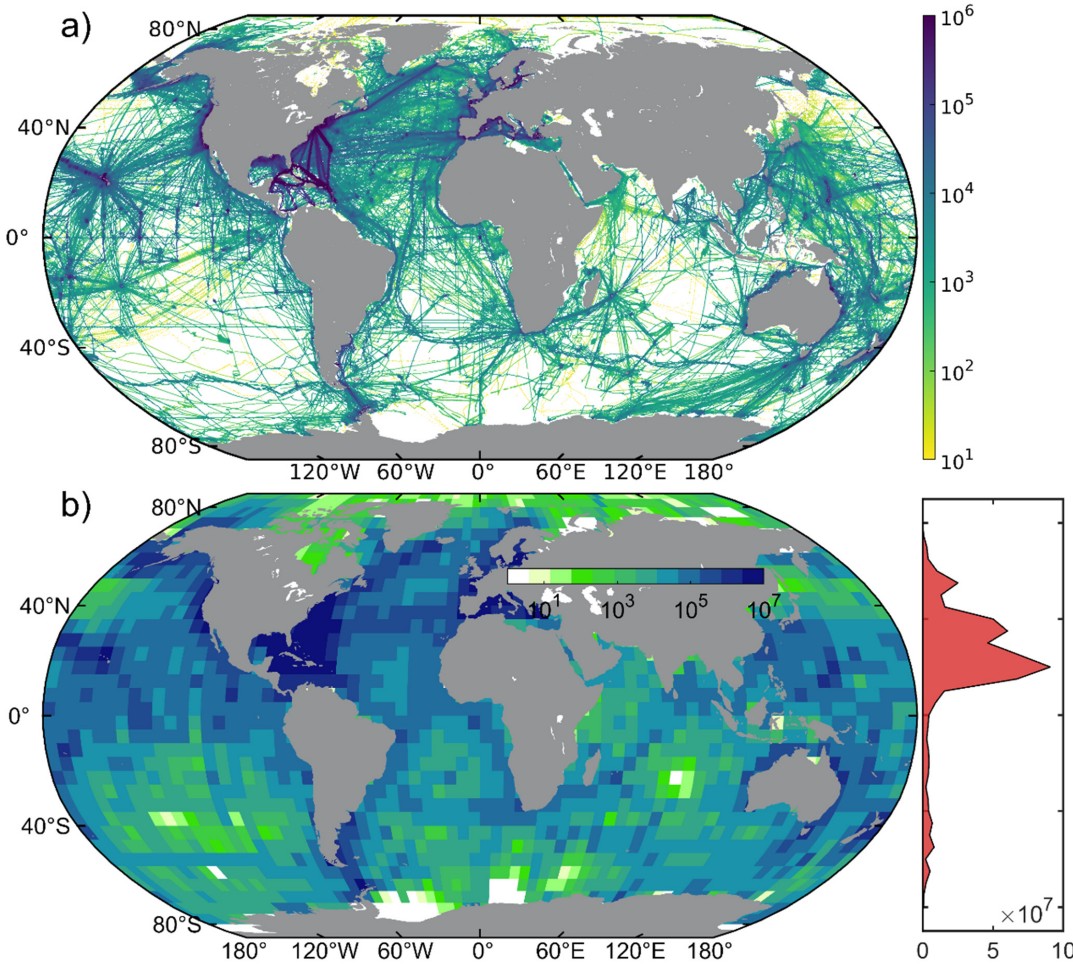

**Figure 1: The data distribution of Surface Underway Marine Database. a) Trajectories of all SST underway observations, with**
**colours indicating the number of data records. b) The number of SST records within each 5° × 5° grid, along with their meridional**
**averages.**

## 3 Methods

### 3.1 Global front detection algorithm

The histogram-based CCA is designed to identify boundaries that separate distinct water masses, defining them as oceanic

fronts (Cayula and Cornillon, 1992; Cayula and Cornillon, 1995). An enhanced version, the CCAIM, was proposed by Xing

et al. (2023a), integrating inverse distance weighting and mathematical morphology operators into the original CCA to better

capture previously missed coastal fronts, significantly improve frontal continuity, and reduce duplicate detections. In this study,

we refined CCAIM by introducing a pre-processing step to enhance its suitability for global front detection and incorporated

additional functionality to identify frontal zones along with their warm and cold sides. The final dataset includes detailed

position information about the fronts and frontal zones, along with indications of their warm and cold sides, and can be accessed

at https://doi.org/10.5281/zenodo.14373832.

### 3.1.1 The procedural steps in CCAIM

This global detection algorithm consists of six key processes: pre-processing, histogram analysis, cohesion testing, frontal localization, multi-window combination, and front pruning. During pre-processing, CCAIM applies inverse distance weighting to estimate SST values along the edges of land and regions contaminated by clouds, using available SST pixels nearby. A 3 × 3 median filter is applied to reduce random noise, and daily SST images are then divided into 32 × 32 pixel windows with 16-pixel overlaps. Within each window, histogram analysis, cohesion testing, and frontal localization are conducted independently. For histogram analysis, a threshold based on the ratio of between-cluster to total variance (set at 0.75 here) assesses whether two distinct SST populations are present. If these populations exceed a cohesion coefficient of 0.93 and have a mean SST difference over 0.25 °C, CCAIM designates edge pixels in the warm SST population adjacent to the cold population as frontal pixels, compiling results from all overlapping windows to form a candidate frontal map (as shown in Fig. 2a). Following this, CCAIM performs a series of morphological operations, such as closing, thinning, and filling, to bridge any fragmented fronts with gaps of < 2 pixels and to create fronts of one-pixel width (Fig. 2b). A contour pruning algorithm is then applied to segment the candidate front branches into independent fronts, removing any branches shorter than 10 pixels (Fig. 2c).

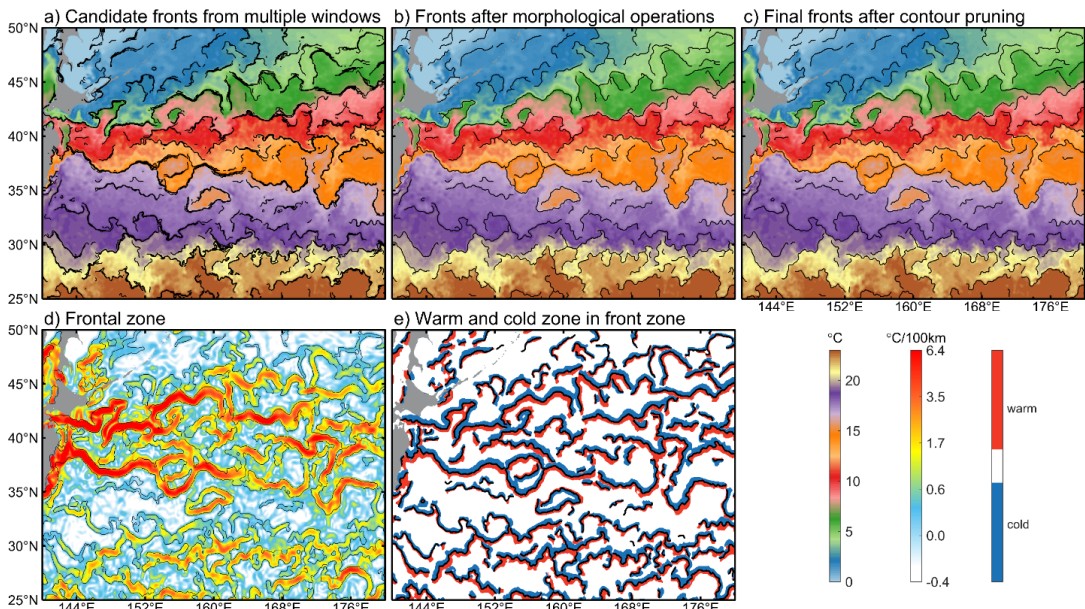

**Figure 2:** Process of the front detection algorithm applied to the SST image from January 25, 2023. The colours of a)–c) denote the SST values, while and d) and e) represent SST gradient field and the identified warm/cold frontal zones.

### 3.1.2 Modifications for global front detection

Due to its large statistical window, CCAIM faces challenges in detecting fronts near image edges. Additionally, fixed-window segmentation in CCAIM often produces artefacts in long-term frontal occurrence maps, as shown in Fig. S1, resulting in visible stripes in both longitudinal and latitudinal directions from detection biases in the fixed window approach. To mitigate these issues, our global detection algorithm introduces some modifications for daily global SST images (Fig. 3). First, we create two duplicated edge zones in the latitudinal direction with a random width of 33–64 pixels and link them to their opposite edges. Second, in the longitudinal direction, we add a land-pixel zone at the image edges near Antarctica and a sea ice-pixel zone at the image edges near the Arctic, both with a random width of 1–32 pixels (Fig. 3). These random-width buffer zones, designed according to the 32-pixel window size of CCAIM, effectively reduce artefacts caused by fixed-window segmentation, thereby enhancing the reliability of front detection.



Identifying frontal zones and their warm and cold sides enables researchers to quantitatively investigate the ecological and climatic effects driven by fronts, such as the differing ecological responses on the warm and cold sides of these zones (Snyder et al., 2017; Zhang et al., 2019). In our global front detection algorithm, mesoscale SST gradient fields are calculated using a modified Sobel operator (Xing et al., 2022; Xing et al., 2023b), followed by a logarithmic transformation to approximate a Gaussian distribution. Pixels within the range where the SST gradient magnitude decreases to half the value of the nearest frontal pixels are designated as frontal zones. This aligns with the definition of a frontal zone as an area where the SST gradient magnitude is relatively higher than that of surrounding waters (Legeckis, 1978). It is important to note that the threshold value used for defining frontal zones is subjective, and users can adopt alternative threshold values that suit their research needs, such as a fixed width around the front or variable threshold values (Xing et al., 2023b). Morphological operations are applied to the candidate frontal zones identified by the aforementioned threshold value. The execution of these operations is similar to those used in the multi-window combination in CCAIM, which aims to reduce anomalous pixel spines and gaps within frontal zones. Subsequently, this algorithm compares the SST values of each pixel within the frontal zones to those of the nearest corresponding frontal pixels, identifying pixels with higher SST as the warm side and those with lower SST as the cold side of the frontal zone. Figures 2d and 2e illustrate examples of the detected frontal zones and their warm/cold sides derived from the SST image.

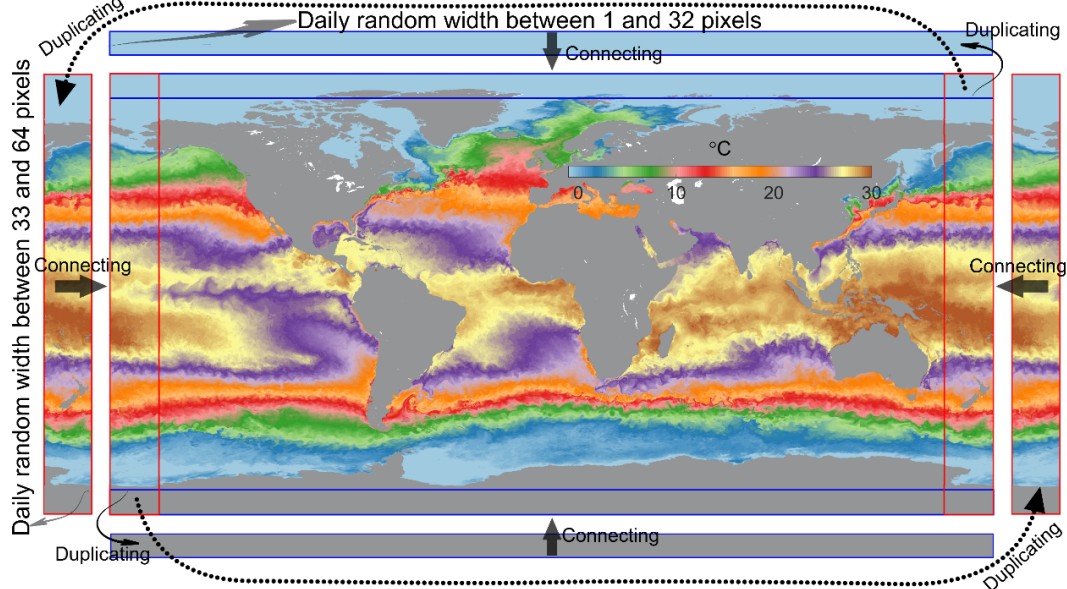

**Figure 3: A pre-processing in image edge for global front detection.**

### 3.2 Front detection from underway data

High-resolution underway observations offer an effective approach for validating the presence of satellite-based fronts along specific trajectories. For this analysis, all underway observations were initially divided into various continuous trajectories, defined by intervals where either the time between two adjacent observations exceeded 0.5 days or the distance exceeded 0.1°. Short trajectories of less than 1° were excluded, and a 0.05° median filter and a 0.05° mean filter were applied to remove anomalous values and submesoscale SST variations. To address spatial resolution discrepancies between underway and satellite data, all continuous SST underway observations were linearly interpolated along their trajectories to a 0.05° spatial resolution matching that of the satellite data. Subsequently, along-track mesoscale SST gradient magnitudes were calculated



by the absolute SST difference at a distance of 0.25° between any two sides of any observed point along the underway trajectories, followed by conversing their unit to °C/100 km. This approach aligns with the modified Sobel operator, which is optimized for quantifying mesoscale SST gradient variations while ignoring submesoscale information (Xing et al., 2022; Xing et al., 2023b). Following definitions from previous studies (Fedorov, 1986; Ullman and Cornillon, 2000), mesoscale

fronts in the SST underway data were identified at observation points where along-track SST gradient magnitudes were local maxima, either exceeding 4 °C/100 km or surpassing 1 °C/100 km while being more than twice the average gradient within a 32-point range along the trajectory. In this process, the selection of a 32-observation-point range is intentionally designed to align with the 32-pixel window size used in CCAIM. Figure 4a shows a typical example of identified fronts from along-track SST data observed by the Saildrone SD-1021 in February 2019.


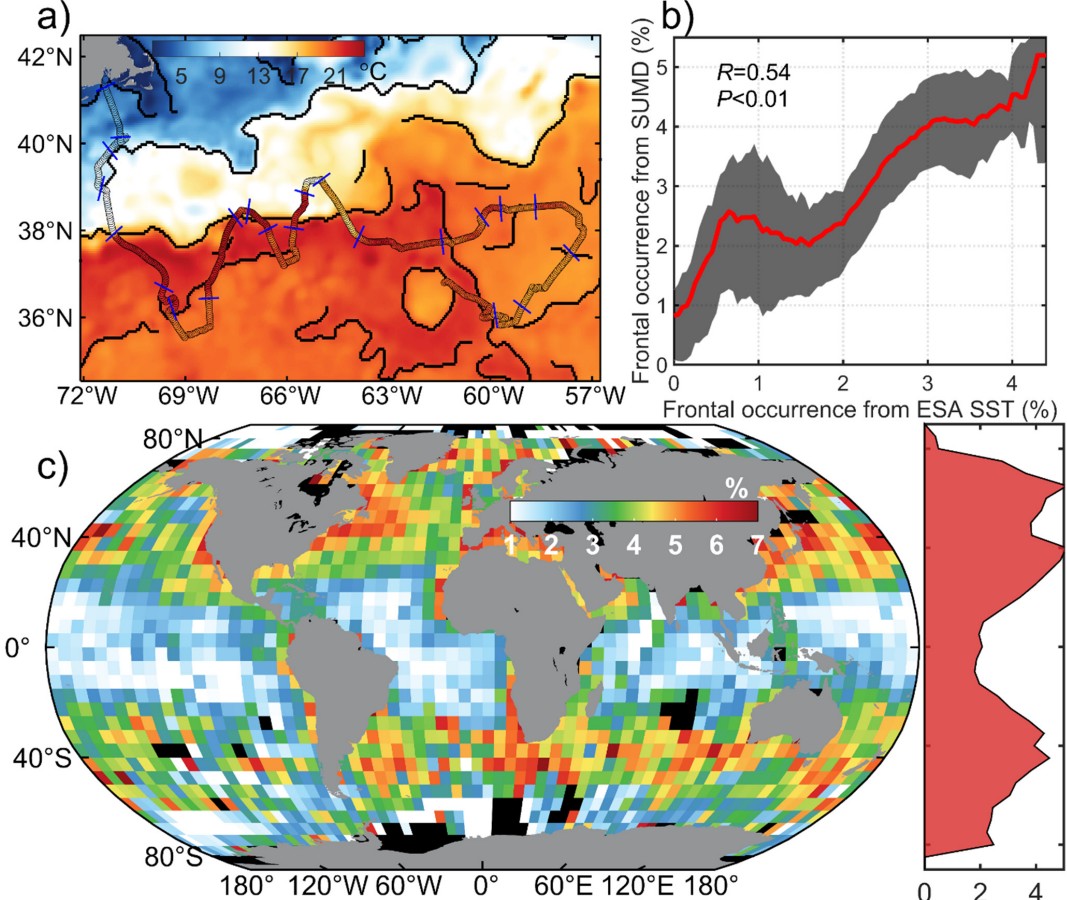

**Figure 4: Ocean fronts identified from underway observations. a) An example of along-track fronts observed by Saildrone SD-1021 from January 1 to February 25, 2019. Coloured points represent underway SST observations, and blue orthogonal lines indicate the locations of detected fronts. The colours correspond to ESA SST data from February 12, 2019, while black lines represent the**
**detected fronts. b) Comparison of global long-term frontal occurrences identified by SUMD data with those detected by ESA data on a 5° × 5° grid. The red line represents the average values of SUMD-based frontal occurrence, while the light-grey shading denotes the 25th and 75th percentiles. c) Long-term frontal occurrence calculated from SUMD-detected fronts (1989–2023).**

### 3.3 Statistical analysis

Although ocean fronts detected from in-situ underway data require a subjective threshold, they provide an independent global
comparison with satellite-based front detection. We hypothesize that the fronts identified from SUMD data are accurate and



use hit rate and precision as two statistical indicators to validate and compare our global front detection across different datasets (Xing et al., 2023a). The hit rate is defined as the proportion of fronts detected from underway data that can be found within the satellite-based fronts within a 3-day window and a 5-pixel (~25 km) radius. Satellite-identified fronts are interpolated to the underway trajectories with a 0.05° spatial resolution using the nearest-neighbour method. Precision is then calculated as

the proportion of satellite-identified along-track front data that can be found within the SUMD-based fronts within a 3-day window and a 5-pixel along-track radius. For fronts detected from the MODIS SST dataset, we extended the window to 8 days to match its 8-day temporal resolution. We calculated these two indicators within each 5° × 5° grid to examine the spatial distribution of satellite-based front detection performance, and computed them for each year and month to explore their interannual and seasonal variations.

The long-term frontal occurrence frequency was also used for cross-dataset comparison to assess the robustness of our global front datasets. For satellite-based front data, this metric is calculated as the ratio of days with detected fronts to total observation days within each pixel. For SUMD-based fronts, it is calculated as the ratio of points with detected fronts to the total number of observation points along trajectories within each 5° × 5° grid. We calculated the average frontal occurrence, along with the 25th and 75th percentiles at 0.05% intervals, followed by a 0.5% moving average filter to reduce noise. We also computed the

Pearson correlation coefficients of the front distribution to compare the relationships between our front datasets and those obtained from other SST datasets. The hit rate and precision of ESA SST-based persistent frontal pixels, calculated within a 30 km range, were also used to compare frontal occurrence with those from other datasets (Xing et al., 2024b).

## 4 Results

### 4.1 Validation of global front dataset using underway data

The hit rate quantifies the proportion of true fronts (SUMD-based fronts) that can be detected through satellite observations. In our global front dataset, hit rates are consistently high across most global ocean regions, with zonal averages typically ranging from 70% to 90% and a global integrated value of 77.65% (Fig. 5a). Low hit rates are observed in the western equatorial Pacific and the southern regions of the Southern Ocean, where both underway observations and satellite-detected fronts are sparse. Significant interannual and seasonal variations in hit rate are also observed, fluctuating between 70% and

85% (Fig. 5b and 5c). A slight decreasing trend appears alongside the rapid increase in observational data, with peak hit rates occurring between August and November, and lower rates typically seen in May and June in the Northern Hemisphere. Overall, the consistently high hit rate across the global oceans indicates that most of the SUMD-detected fronts can be reliably identified by our global front dataset.

Precision quantifies the proportion of true fronts (SUMD-based fronts) among all satellite-detected fronts. The precision of

our global front dataset exhibits significant meridional variations, with a global integrated value of 67.43%. It peaks in temperate waters at approximately 70% and reaches its lowest value around equatorial waters at about 35% (Fig. 6a). Higher precision is observed in strong-front regions, such as coastal and western boundary current waters. When weak fronts (SST gradient < 1.5 °C/100 km) are excluded, precision consistently shows higher values across the global oceans, reaching 74.69% (Fig. 6b). The low precision in equatorial waters becomes negligible after excluding weak fronts, as weak fronts are particularly

prominent in this region. Precision shows a slight increasing trend with the rise in observational fronts and exhibits large fluctuations prior to 2001 due to limited data availability, stabilizing after 2001. Seasonal variations in precision follow a dome-shaped pattern, peaking during warm periods and bottoming out during cold periods. Overall, the global front dataset captures more weak fronts that cannot be detected by SUMD, while strong fronts identified by SUMD and ESA SST datasets show better consistency.




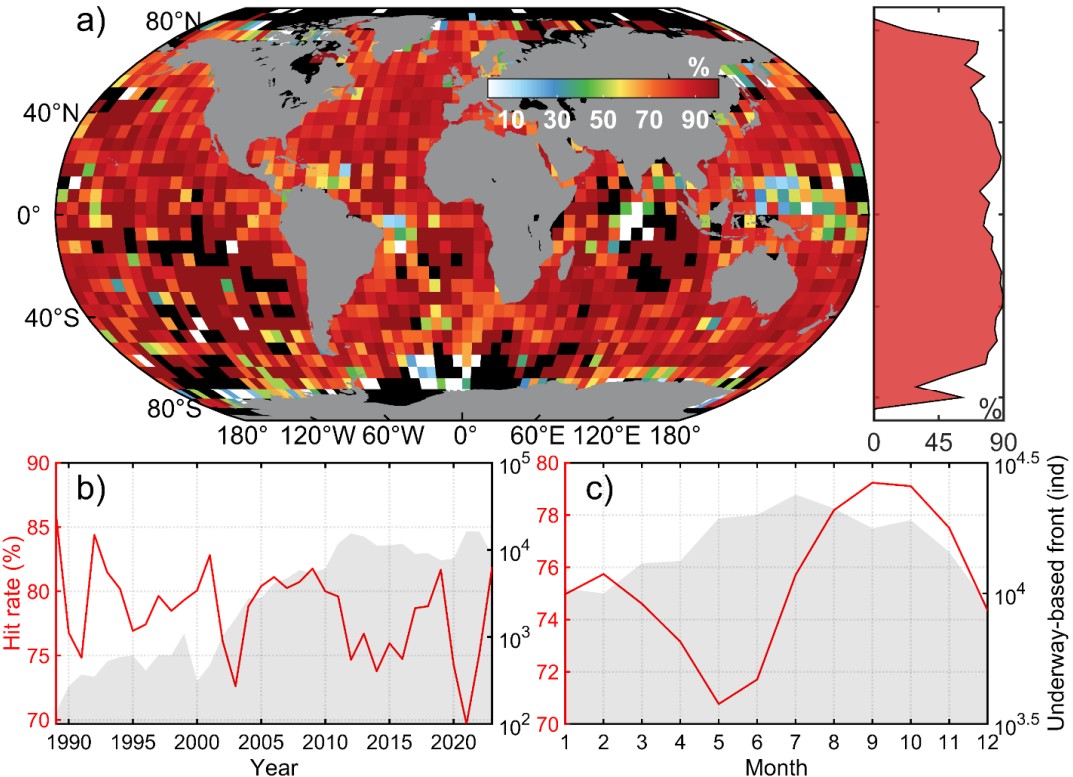

**Figure 5: Hit rate of fronts detected from ESA SST. a) Spatial distribution of hit rates and their zonal averages from 1989 to 2023. Black regions indicate areas with no frontal data. b) and c) Annual (b) and monthly (c) hit rate across global oceans. The months in the Southern Hemisphere were converted to the Northern Hemisphere by adding 6 months.**




Earth System
Science
Data

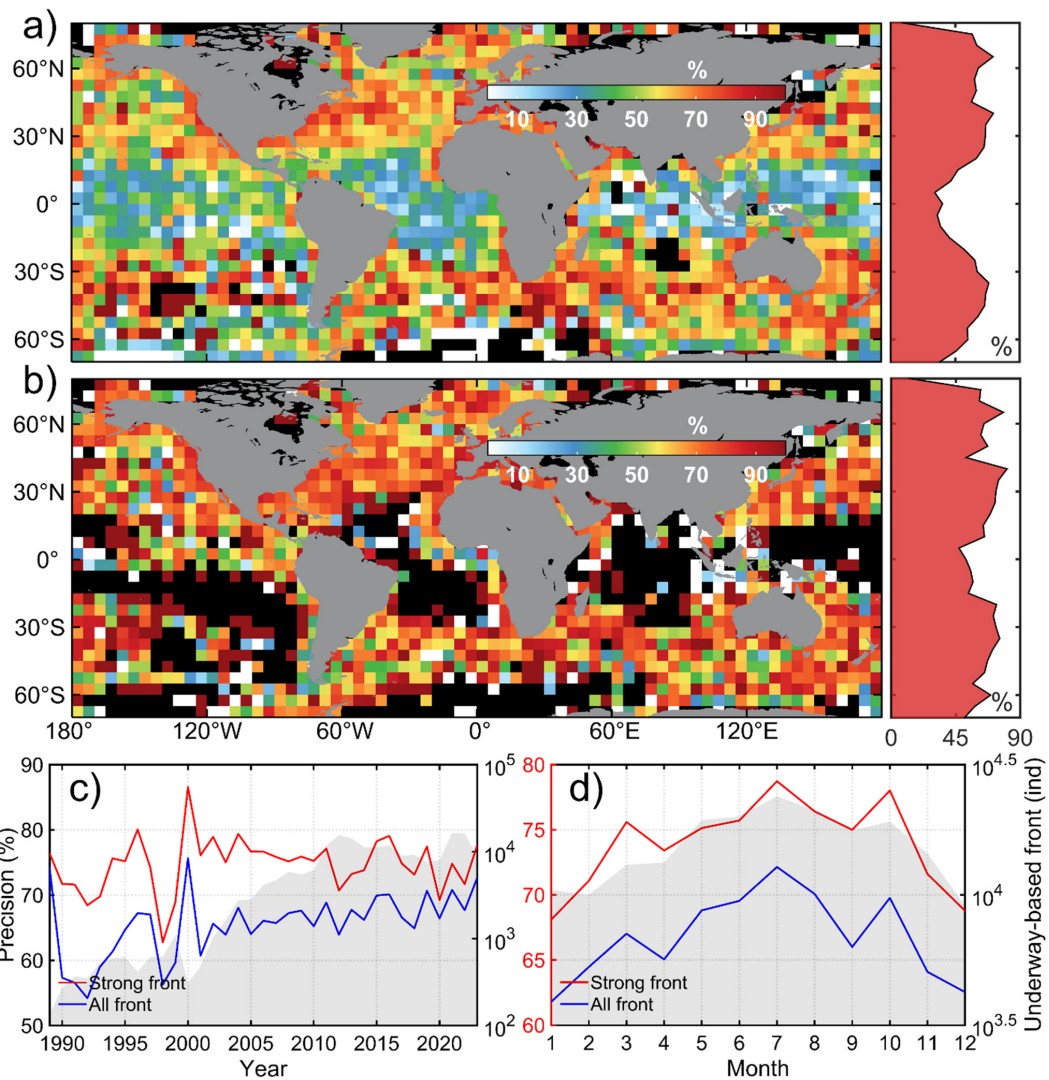

**Figure 6: Precision of fronts detected from ESA SST. a) and b) Spatial distribution of precision and their zonal averages for all fronts (a) and strong fronts (b) from 1989 to 2023. Black regions indicate areas with no frontal data. c) and d) Annual (c) and monthly (d) precision across global oceans. Strong fronts represent the fronts with >1.5 °C/100 km SST gradient.**

### 4.2 Cross-dataset comparison with underway data

Statistical indicators for global front detection show significant variation across different SST datasets (Fig. 7). REMSS and ESA-identified fronts exhibit similar spatial patterns in hit rate and precision (Fig. S2), achieving the highest values across the datasets, exceeding 75% in hit rate and 60% in precision. In contrast, fronts detected using the MODIS dataset show the lowest hit rate at only 50.98%, suggesting a strong impact of cloud contamination on front detection accuracy. Additionally, MODIS SST data usually show higher error rates at high latitudes, which likely contributes to the lower hit rate in these regions (Fig. S2a). The hit rate for MODIS data improves only marginally to 51.62% after excluding data north of 70°N and south of 60°S, underscoring MODIS's relatively poor performance in front detection compared with other datasets. GLORYS and HadGEM3 show moderate global hit rates but the lowest precision, reaching only 56.83% and 58%, respectively (Fig. 7). GLORYS benefits from extensive observational assimilation, achieving a hit rate 7.1% higher than the purely simulated HadGEM3, reflecting the added value of assimilated observations in numerical simulations for front detection.

Although GLORYS has a lower hit rate, its spatial pattern aligns closely with those of ESA and REMSS (Figs. 5a and S2). For MODIS and HadGEM3-detected fronts, lower hit rates are observed in equatorial waters. The spatial distribution of precision across datasets mirrors that of ESA-detected fronts, with higher precision typically found in strong-front regions and lower precision around the equator (Fig. S2). Similar to ESA SST, a slight upward trend in hit rate is also seen in MODIS and

REMSS-detected fronts, ranging from 70% to 85% and 45% to 60%, respectively. In contrast, hit rates for GLORYS and HadGEM3-detected fronts remain stable despite the sharp increase in available observations. Precision shows relatively low variation but presents an upward trend, which can be attributed to the increase in available observations. Generally, multi-satellite merged REMSS and ESA SST products present the best front detection performance among the five datasets, followed by observation-assimilated GLORYS product, while single-satellite MODIS and purely simulated HadGEM3 have the lowest

detection performance.

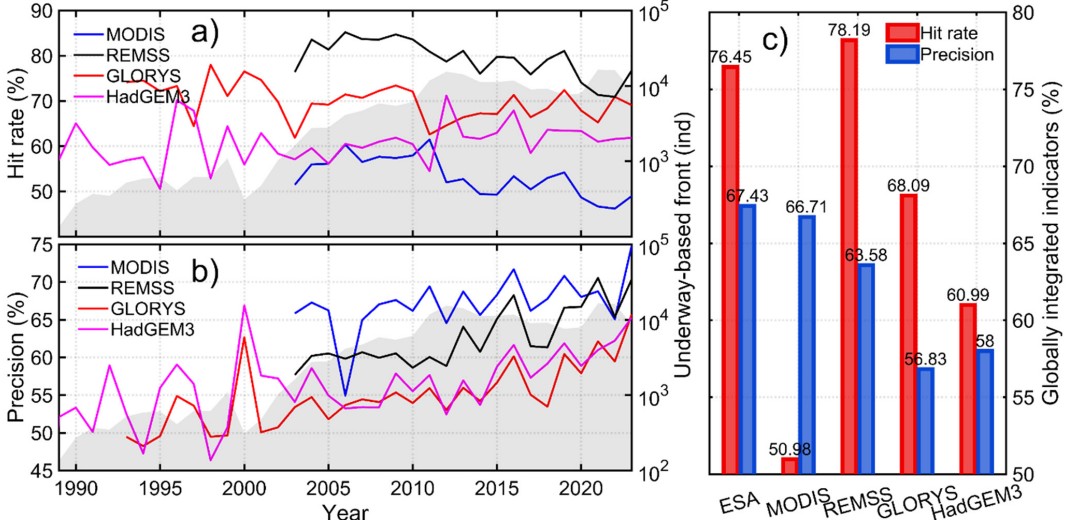

**Figure 7: Cross-datasets of frontal statistical indicators. a) and b) Hit rate (a) and precision (b) of fronts obtained from different datasets based on SUMD-detected fronts. a) Globally integrated hit rate and precision based on SUMD data.**

**4.3 Cross-dataset comparison of long-term frontal occurrence**

Figure 8 illustrates the global spatial distribution of frontal occurrence frequency based on ESA SST front detection data over the past 42 years. High frontal occurrence frequency is found in coastal waters, boundary currents, and the Antarctic Circumpolar Current, whereas equatorial waters and the interiors of subtropical gyres exhibit the lowest frequencies. Additionally, frontal occurrence peaks in temperate regions, largely due to the presence of western boundary currents and their

extensions, while reaching a minimum in tropical regions. This spatial distribution closely resembles that derived from SUMD data, with SUMD-based frontal occurrence increasing in tandem with ESA-based frontal occurrence, showing a correlation coefficient of 0.54 at the 0.01 significance level (Fig. 4).

Frontal occurrences derived from other satellite- and simulation-based datasets also show significant correlations ($P < 0.01$) with those from our global front dataset (Figs. 9 and 10). MODIS SST-derived frontal occurrence is notably low, likely due to

cloud-related data gaps and elevated noise in the MODIS dataset (Fig. 9b). REMSS SST-derived fronts display the highest correlation with ESA SST; however, significant artefacts appear in the long-term frontal occurrence data, with stripes of high occurrence frequency evident in temperate waters, despite the use of random windows in front detection (Fig. 9c). MODIS and REMSS SST-derived frontal occurrences have the highest hit rate but lowest precision, indicating that they capture more persistent frontal pixels than ESA SST, potentially due to noise-related frequency variations and artefacts of stripes (Fig. 9e).

For GLORYS, frontal occurrences are relatively high in areas with 1%–4% frequency, closely aligning with ESA SST data in regions above 4% and showing both a high hit rate and precision (Fig. 10), thus demonstrating strong similarity to ESA SST fronts. In contrast, HadGEM3 SST-derived frontal occurrences show the lowest similarity to ESA SST, with a relatively low occurrence frequency and precision. Overall, cross-dataset comparisons support the robustness of the long-term frontal occurrence and the detection of persistent fronts in our global front dataset.

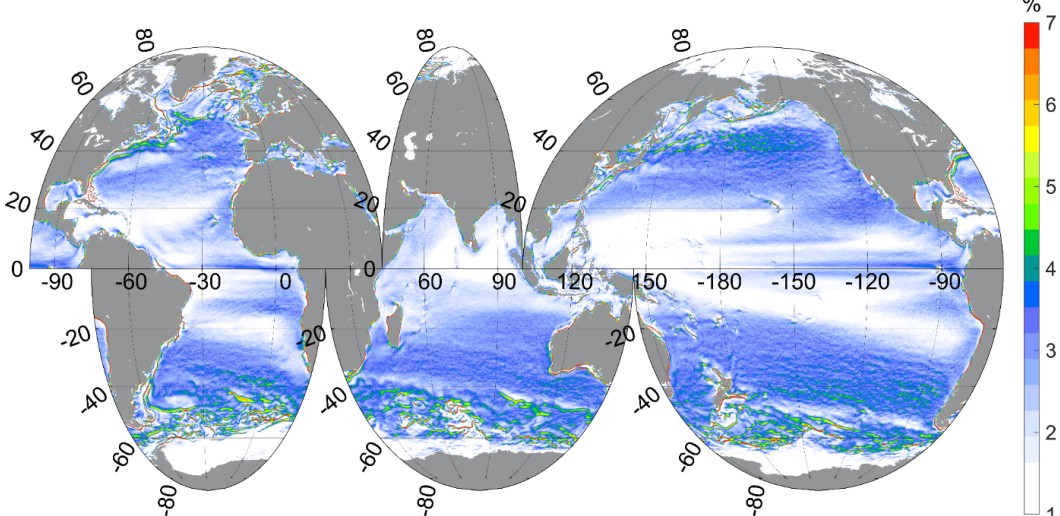


**Figure 8: Frontal occurrence frequency detected from ESA SST from 1982 to 2023.**

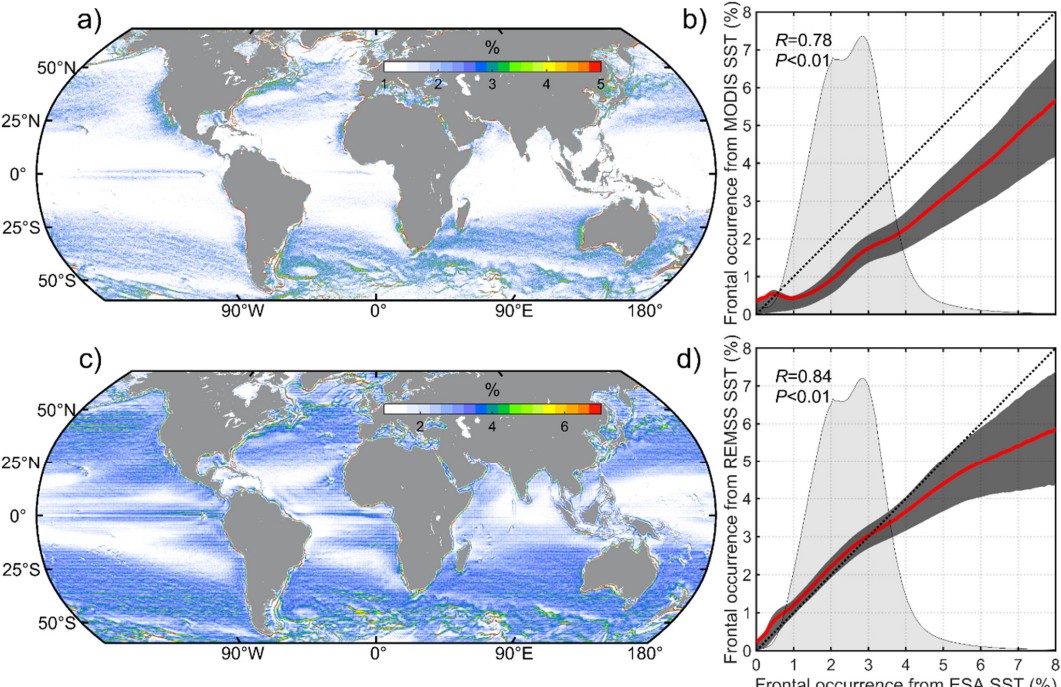

**Figure 9: Frontal occurrence detected from MODIS (a) and REMSS (c) SST datasets from 2003 to 2023. The relationship of frontal occurrence detected from MODIS (b) and REMSS (d) and those from ESA SST. The red line represents the average values of**
**SUMD-based frontal occurrence, with light-grey shading indicating the 25th and 75th percentiles. The dotted lines represent the fitted lines where frontal occurrences from the two products are equal, and the grey shading indicates the normalized count of available data.**

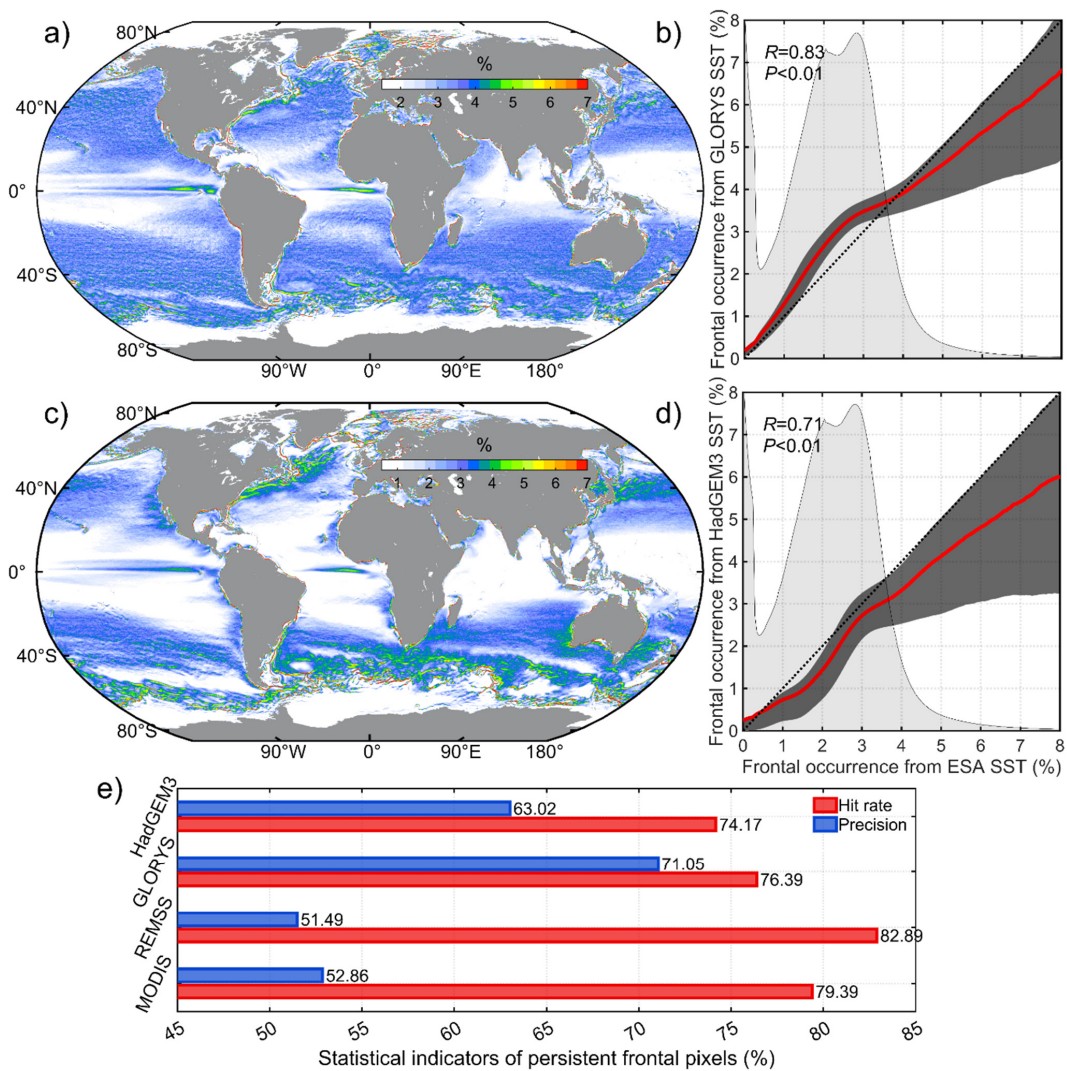

**Figure 10: Same as Fig. 9, but for GLORYS (a, b) and HadGEM3-detected (c, d) fronts. e) cross-dataset statistical indicators of persistent frontal pixels.**


## 5 Discussions

Although ocean fronts have increasingly attracted the interest of researchers due to their broad impact on the oceanic realm, these oceanographic features are often implicitly represented in both observed and simulated ocean data (Belkin, 2021). This

creates a gap in interdisciplinary research, compelling oceanographers and ecologists to invest significant time and resources in obtaining fundamental frontal data. This study provides the first publicly available global ocean front dataset covering the past 42 years, with broad applications in oceanography, ecology, and fisheries research. Climate change is altering global oceanic circulation and dynamic processes (Cheng et al., 2022), potentially leading to significant changes in the occurrence, intensity, and position of ocean fronts (Franco et al., 2022; Kahru et al., 2018; Xing et al., 2024b). Our frontal dataset offers a

valuable tool to investigate changes in fronts and their impacts on the redistribution of ecological and climatic services under





global warming. Given the importance of ocean fronts in stimulating marine productivity and attracting marine organisms, this daily frontal information can be easily integrated with biogeochemical data, plankton data, and fish and marine predator distribution data (Haëck et al., 2023; Miller et al., 2015; Nieto et al., 2017). This will help quantify front-induced ecological impacts and elucidate the underlying biophysical coupling mechanisms, supporting more accurate ecological predictions. More

importantly, given the different ecological roles of the warm and cold sides of fronts (Snyder et al., 2017; Zhang et al., 2019), our dataset also provides information on the warm and cold sides within frontal zones, facilitating statistical investigations similar to the comparative analyses between cyclonic and anticyclonic eddies (Chelton et al., 2011; Xing et al., 2023c). Additionally, our front data offer valuable references for the design of marine protected areas, considering the potential aggregation effects of human activities, protected species, and floating marine debris (Le et al., 2024; Miller et al., 2015;

Queiroz et al., 2016). With the significant progress in artificial intelligence models, deep learning methods have been developed to detect and predict ocean fronts (Yang et al., 2024). However, obtaining a reliable training set for front occurrence remains a challenge. Our global front dataset fills this gap and may facilitate further research in this field.

Satellite-based automatic front detection algorithms typically rely on relatively subjective threshold values, necessitating comprehensive validation before application (Belkin et al., 2023; Ullman and Cornillon, 2000; Yao et al., 2012). However,

obtaining large-scale in-situ observations to validate global frontal detection remains challenging, resulting in prior studies primarily focusing on a limited number of known fronts or those within specific regions (Chang and Cornillon, 2015; Miller, 2009). For the first time, this study is the first to utilize global underway observations for validating the CCAIM on a worldwide scale. The spatial distribution of validated results demonstrates a globally consistent high performance, with 77.65% of fronts accurately identified by CCAIM and a precision of 67.43%. These results align with the findings of Ullman and Cornillon

(2000), who reported missed front error rates (1 - hit rate) and false front error rates (1 - precision) of approximately 30% and 27%–28%, respectively, for CCA in the North Atlantic Ocean. Satellite SST images are captured at fixed times each day, which can differ from the timing of underway data collected on the same day. This temporal discrepancy may introduce variability due to intraday front and SST changes, affecting statistical indicators. Our global front detection using CCAIM excluded short fronts and branches shorter than 10 pixels, which are often identified by SUMD algorithms, potentially lowering

the hit rate. The histogram-based algorithm identifies fronts by distinguishing between two water masses within a single window (Chang and Cornillon, 2015). However, when multiple water masses are present within the same window, it may fail to detect some fronts, partially explaining the reduced hit rate. Additionally, the large window size of the histogram-based algorithm, combined with the morphological operator applied to multiple window combinations, is designed to identify longer, spatially continuous fronts (Chang and Cornillon, 2015; Nieto et al., 2012; Xing et al., 2023a). In some cases, this approach

may generate fronts in regions without local maxima in the SST gradient by connecting fragmented surrounding fronts (Cayula and Cornillon, 1992; Xing et al., 2023a). In contrast, SUMD-based SST front detection does not consider the spatial distribution of SST, potentially overlooking certain fronts due to disturbances from local submesoscale processes or anomalous surface observations, thereby reducing precision values.

Notably, precision improved to 74.69% when weak fronts with SST gradients <1.5 °C/100 km were excluded. This aligns with

previous findings by Ullman and Cornillon (2000) and Chang and Cornillon (2015), which reported that false front error rates significantly decreased as SST gradients increased. Fronts in equatorial waters typically exhibit weak SST gradients due to uniform heating from intense solar shortwave radiation (Xing et al., 2023b). Consequently, fewer "true" fronts can be detected in SUMD-based in-situ observations due to its fixed gradient thresholds, resulting in relatively lower precision in our dataset and a potential risk of identifying more "false" fronts. For gradient-based frontal detection algorithms, Xing et al. (2023b)

recommend using a sliding-window threshold to improve the detection of weak fronts in low-gradient regions and seasons. Adopting these variable thresholds in SUMD-based frontal detection could potentially enhance the precision values of our global front dataset significantly. However, SUMD-based frontal detection also relies on relatively subjective thresholds due to the lack of explicit frontal definition (Fedorov, 1986), even though both previous studies and our validation assume the



detected features to represent "true" fronts (Chang and Cornillon, 2015). Using higher gradient thresholds generally identifies
fewer fronts, which increases the hit rate while lowering precision, and vice versa. To support the robustness of our validation,
we conducted a sensitivity analysis on the thresholds used in SUMD-based frontal detection. Specifically, we adjusted the "2
times" criterion in the "more than 2 times the average gradient" in the Method section to 1, 1.5, 2.5, and 3 times. The results
showed minimal changes in the hit rate, while precision decreased with higher thresholds (Table 2), further demonstrating the
reliability of our global validation. Overall, the statistical indicators of our dataset can be considered to demonstrate acceptable
performance for accurately tracking front occurrence (Ullman and Cornillon, 2000).

**Table 2: The sensitivity analysis of the thresholds used in SUMD-based frontal detection. The "2 times" threshold in the criterion "more than 2 times the average gradient" in the Methods section was adjusted to 1, 1.5, 2.5, and 3 times for sensitivity analysis.**

| Statistical indicators | 1 | 1.5 | 2 | 2.5 | 3 |
|---|---|---|---|---|---|
| Hit rate | 76.90% | 77.58% | 77.65% | 78.87% | 78.81% |
| Precision | 71.13% | 69.70% | 67.43% | 64.14% | 61.11% |

SST datasets have been widely used in studies of front detection, each with its own strengths and limitations. Multi-satellite
merged SST data effectively reduce the impact of cloud-induced missing values (Embury et al., 2024); however, their frontal
detection can be affected by errors introduced during the temporal and spatial merging of data from multiple sources.
Conversely, single-satellite SST observations avoid the influence of multi-source data fusion but are criticized for the
significant number of missing values due to cloud contamination (Suberg et al., 2019). Similarly, numerical simulations of
SST may generate false fronts due to model-related uncertainties (Jean-Michel et al., 2021). Our results revealed that cloud-
contaminated SST datasets, such as MODIS, missed approximately half of the SUMD-based in-situ fronts. In contrast, multi-
satellite merged SST products like ESA and REMSS showed the best performance among all datasets analysed (Fig. 7). Ullman
et al. (2007) noted that frontal occurrence derived from cloud-contaminated images was half that of cloud-free images, and
Obenour (2013) suggested that at least 90% of available SST data within the CCA windows is necessary to produce reliable
frontal occurrence estimates. Despite the application of cloud-contaminated SST datasets in assessing long-term frontal
occurrence changes under global warming (Yang et al., 2023), Suberg et al. (2019) indicated that missing values caused by
clouds and sea fog could introduce spurious trends. No abrupt changes in frontal occurrence were observed in the validation
of multi-satellite merged SST datasets over the past 30 years, both temporally and spatially (Figs. 5 and 7). This consistency
suggests that the processes of multi-source data fusion do not significantly affect their spatial and temporal variations, making
these datasets more suitable for statistical studies on frontal occurrence changes. Furthermore, our comparison of validation
results between observation-assimilated ocean models and pure ocean models suggests that observation assimilation enhances
the accuracy of ocean front simulations. However, the relatively low performance of these models highlights the need to
improve their ability to simulate mesoscale and submesoscale fronts (Roberts et al., 2019). Enhancing this capability is
essential for optimizing Earth system models and improving the accuracy of climate projections, given the critical role these
processes play in air-ocean interactions (D'Asaro et al., 2011).

Clarifying the global pattern of frontal occurrence is essential for investigating spatiotemporal variations and identifying
persistent oceanic fronts. Recently, several studies have proposed global frontal occurrence maps using different detection
algorithms and SST datasets (Mauzole, 2022; Xing et al., 2023b; Xing et al., 2024b). However, these global patterns rely solely
on satellite observations and lack validation against independent in-situ observations. Our comparisons between ESA-based
and SUMD-based fronts indicate that satellite-based automatic front detection reliably captures the primary spatial pattern of
frontal occurrence. Notable features, such as more frequent frontal occurrences in coastal regions, boundary current and
extension regions, as well as Antarctic Circumpolar Current regions, are consistently observed in the SUMD-based frontal
occurrence map (Fig. 4). Despite the general consistency across maps, some well-known persistent fronts in open oceans, such





as boundary current fronts and Antarctic Circumpolar Current fronts, are absent in maps of Mauzole (2022). Additionally,
many shelf and slope fronts appear less discernible or disappear entirely in Mauzole's maps but are clearly visible in other
maps. Whether this discrepancy arises from advancements in detection algorithms or differences in SST datasets remains
unclear (Xing et al., 2023b). Our comparisons reveal that the occurrence frequency in MODIS maps is lower than that in maps
derived from multi-satellite merged ESA and REMSS SST products due to missing values caused by cloud and sea fog (Figs.
9 and 10). However, even cloud-contaminated MODIS SST data can produce detailed maps highlighting well-known persistent
fronts in both coastal and open oceans, which are not adequately captured in maps of Mauzole (2022) despite using similarly
cloud-contaminated SST images. This demonstrates that improvements in frontal occurrence maps from Xing et al. (2024b)
are largely attributed to advancements in detection algorithms rather than solely addressing missing values. Meanwhile, many
anomalous values due to missing data lower the precision of persistent frontal pixels in MODIS-derived products. For REMSS-
based frontal occurrence maps, artefacts in the form of stripes with a 2° interval, corresponding to the operational windows in
REMSS's optimal interpolation process for merging multi-satellite observations (Kawai et al., 2006), reduce precision. Further
investigation into the relationship between these artefacts and the merging process may enhance the robustness of REMSS
products in depicting frontal variations. Overall, among these products, the ESA SST product used in our global front dataset
appears to be the most suitable for studying frontal changes and conducting statistical analyses.

**6 Data and code availability**

The code and data of our global daily mesoscale front dataset can be accessed at https://doi.org/10.5281/zenodo.14373832
(Xing et al., 2024a). Our global front dataset from 1982 to 2023 is archived in NetCDF format under the variable name "front",
comprising 15,340 individual daily files. The daily "front" variable is stored as a two-dimensional numerical matrix with
dimensions of 7200 in the zonal direction and 3600 in the meridional direction. The values of the "front" variable are unitless,
where regions with values greater than 0 represent the warm sides of detected frontal zones, while those with values less than
0 indicate the cold sides. All positions where the "front" variable equals -10, 10, or 30 correspond to detected frontal lines,
with these values respectively indicating frontal lines near cold sides, warm sides, and no-front zones. Users can extract specific
frontal properties, such as frontal lines, frontal zones, warm sides, and cold sides, from the "front" variable based on their
interests. Using the detected fronts from January 5, 2023, stored in the NetCDF file 20230105.nc (available at
https://doi.org/10.5281/zenodo.14373832) as an example, Figure 1a displays the detected frontal lines across the global oceans
and the Western North Pacific, plotted using all points where the "front" variable equals -10, 10, or 30. Figure 1b illustrates
the detected frontal zones, highlighting both cold and warm sides. The cold (warm) sides are represented by areas where the
"front" variable is less (greater) than 0, while the contours of the frontal zones are plotted by the boundary lines separating
these regions from the no-front areas.

Earth System
Science
Data

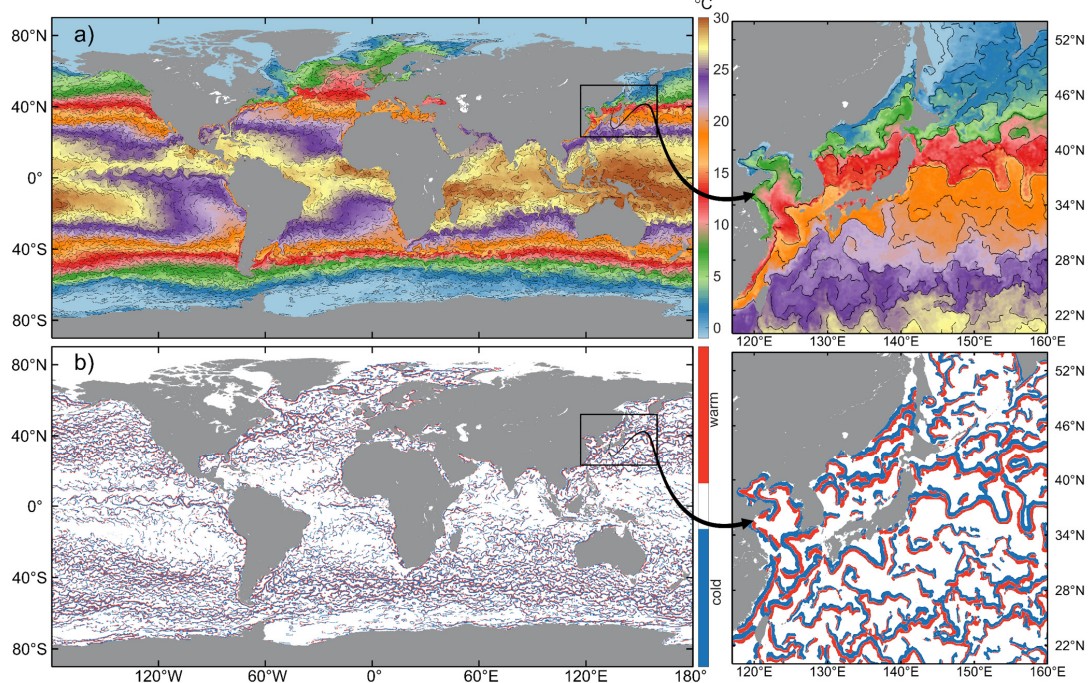

**Figure 11: An example of the global front dataset from January 5, 2023. a) displays the detected frontal lines, while b) illustrates the frontal zones with cold and warm sides.**

## 7 Conclusions

This study extended the recently improved front detection algorithm to global oceans with simple modifications and additional enhancements, such as the use of random windows to reduce artefacts caused by fixed-window segmentation, and the identification of the cold and warm sides of the frontal zone to increase its applicability. This algorithm was then applied to produce the first publicly available daily global ocean front dataset spanning 1982 to 2023 based on 15340 high-resolution SST images (Xing et al., 2024a, https://doi.org/10.5281/zenodo.14373832). Utilizing global in-situ underway observations, this study conducted comprehensive validations and cross-dataset comparisons of satellite-based global front detection.

(1) Statistical indicators show that most of the in-situ observed fronts (77.65%) can be reliably identified by our global front dataset, with high temporal and spatial consistency. Meanwhile, 67.43% of the fronts in our dataset can be matched with in-situ observed fronts, and this value increases to 74.69% if weak fronts are excluded. Given the limitations of underway observations and the gradient-based detection method for fronts, our front dataset demonstrates acceptable performance for accurately quantifying front occurrence.

(2) Cross-dataset comparisons reveal that multi-satellite merged ESA and REMSS products deliver the best front detection performance, followed by the observation-assimilated GLORYS product. In contrast, single-satellite MODIS and the purely simulated HadGEM3 demonstrate the lowest detection performance, primarily due to cloud-related data gaps and simulation inaccuracies.

(3) In-situ observed fronts exhibit a strong spatial resemblance to the long-term global frontal occurrence frequency, providing independent validation of the satellite-based global frontal occurrence map. The high spatial congruence among maps derived from various satellite observations and ocean models further strengthens the robustness of our frontal occurrence pattern and



the detection of persistent fronts. This also suggests that advancements in detection algorithms are a key factor explaining the differences in global front occurrence patterns reported in previous studies.

The comprehensive validation of front detection algorithms using in-situ observations enhances confidence in the application
of satellite-based front detection. Our open-access dataset and detection algorithm are expected to be widely utilized in studies on ocean dynamics, marine ecology, biogeochemistry, ocean management, climate change, and as a training dataset for artificial intelligence in both regional and global oceans.

**Author contributions**

QX developed the method and generated the global front dataset. HY, WY, and XC conceptualized the work and contributed
to data processing. QX and HY created and edited the main figures and wrote the manuscript, while WY, XC, and HW provided valuable insights and made significant revisions to the manuscript. All authors contributed to improving the final version of the manuscript.

**Competing interests**

The contact author has declared that none of the authors has any competing interests.

**Acknowledgements**

We thank the European Space Agency, National Aeronautics and Space Administration, National Oceanic and Atmospheric Administration, Copernicus Marine Service, Remote Sensing Systems, and World Climate Research Programme for their public datasets.

**Financial support**

This research was sponsored by the Shanghai Rising-Star Cultivation Program (Sailing Program) (24YF2716700), the China Postdoctoral Science Foundation under Grant Number 2024M761926, and Shandong Provincial Natural Science Foundation (No. ZR2022QD041).

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
