# Peer review of "A global daily mesoscale front dataset from satellite observations: In situ validation and cross-dataset comparison"

_Earth System Science Data, 2024_

## Referee Comment (RC2)

**Revision of manuscript essd-2024-592**

This paper introduces a valuable global ocean front dataset, employing a refined Cayula-Cornillon algorithm (CCAIM), to provide a daily global high-resolution dataset spanning from 1982 to 2023. The manuscript is well-structured, clearly written, and addresses an important gap in oceanographic research. It also offers rigorous validation against in-situ data and a comprehensive comparison with other datasets. The open access to data and code is commendable and will greatly benefit the scientific community.

Overall, I recommend the manuscript for publication after minor revisions, which are provided below.

- There are frequent repetitions of similar concepts, particularly in the Introduction and Discussion sections. I suggest carefully revising these sections to minimize repetition, which would significantly improve readability.
- The manuscript uses the term "multi-satellite merged product" when referring to gap-free (Level 4) SST products. However, in satellite data terminology, "merged multi-sensor" typically refers to Level 3S (super-collated) products. I suggest using either "multi-satellite blended product," "gap-free," "L4 product," or simply "analysis."
- Lines 24–25: The sentence "...all of which provide independent validation..." is unclear. Please rephrase to improve clarity.
- Line 37: Replace "...affect global climate change..." with "...contribute to climate change...".
- Line 42: The sentence fragment "...ocean fronts oceanographers, climatologists..." seems incomplete or missing an adverb or preposition. Please revise accordingly.
- Line 47: Clarify the term "new technology." Are you referring to the algorithm? If so, consider replacing "technology" with a more specific term, such as "methodology" or "approach."
- Line 58: Replace "accessibility" with "availability."
- Line 106: Clarify "different types." Types of what?
- Line 130: The phrase "serving as a comparison..." is redundant, as this point has already been mentioned. Consider removing or rephrasing.
- Figure 2 caption: There is a typographical error: "while and d) and e)" should be corrected to remove the extra "and."
- Lines 189–190: The sentence starting with "Identifying frontal zones..." is repetitive and not strictly relevant in this section. I recommend removing it.
- Figure 5: The grey shaded area is not described in the caption. Please clarify its meaning explicitly.
- Figure 7: I recommend including the ESA CCI hit rate and precision curves here as well, even though they appear in Figure 5, for immediate comparison.
- Figures 8a, 9a/c, and 10a/c: Figure 8a employs a different map projection from Figures 9 and 10. Please use a consistent map projection to facilitate visual comparisons.
- Lines 343–344: The sentence starting with "Although ocean fronts..." is unclear. Please rephrase to improve readability.
- Line 491: Replace "expected to be widely used" with e.g. "can provide a valuable tool for..."

---

## Author Comment (AC1)

[Figure]

**Figure 3: A pre-processing in image edge for global front detection.**

[Figure]

**Figure 4: Ocean fronts identified from underway observations. a)** An example of along-track fronts observed by Saildrone SD-1021 from January 1 to February 25, 2019. Coloured points represent underway SST observations, and blue orthogonal lines indicate the locations of detected fronts. The colours correspond to ESA SST data from February 12, 2019, while black lines represent the detected fronts. **b)** Comparison of global long-term frontal occurrences identified by SUMD data with those detected by ESA data on a 5° × 5° grid. The red line represents the average values of SUMD-based frontal occurrence, while the light-grey shading denotes the 25th and 75th percentiles. **c)** Long-term frontal occurrence calculated from SUMD-detected fronts (1989–2023).

---

## Author Comment (AC2)

[Figure]

**Figure 7: Cross-datasets of frontal statistical indicators. a) and b) Hit rate (a) and precision (b) of fronts obtained from different datasets based on SUMD-detected fronts. a) Globally integrated hit rate and precision based on SUMD data.**

[Figure]

**Figure 8: Frontal occurrence frequency detected from ESA SST from 1982 to 2023.**

---

## Author Response (AR1)

We sincerely appreciate the editors and reviewers for their time and effort in reviewing our manuscript and providing valuable feedback. We have carefully revised the manuscript in accordance with their suggestions.

**Reply to Referee #1**

The paper presents a valuable public, high-resolution, daily, global front dataset covering the last four decades (1982-2023). The dataset is obtained through the use of an improved histogram-based front detection algorithm and is validated through the comparison with independent in-situ underway observations, demonstrating its good performance in detecting fronts. Developing and validating techniques that enhance our ability to automatically extract frontal information from satellite observations is certainly of great interest to the scientific community and help our understanding of the biophysical interactions within marine ecosystems.

Here, the authors have done an excellent job testing the proposed methodology versus in situ observations and providing a rich cross-dataset comparison to several reanalysis and multi-satellite products, also discussing its advantages and limitations.

Overall, the manuscript is well-written and sections are complete and appropriate. The figures are clear, informative and effectively support the text. Although I have a few concerns about the choice of reference in situ datasets (please, see my comments below), the background, data and methodologies are described well in the introduction and methods sections. A few details are omitted but the authors make reference to a previously published methodology paper to complete information. The results are interesting and well-supported by the figures and a rich discussion. The conclusions provide a concise and accurate summary of the manuscript.

Hence, I recommend publication after the authors have addressed the suggestions and specific remarks reported in the line-by-line comments below:

A: We sincerely appreciate your time and effort in reviewing our manuscript and providing valuable feedback. We have carefully considered all of your suggestions and incorporated them into our revisions.

Q1: Lines 42 and 47: Can you avoid the repetitive use of the word "elucidate", please? A: Thanks for your suggestion. We have changed "elucidate" to "clarify" in Line 43.

Q2: Line 63: Please avoid terms such as "to our knowledge"

A: Thanks for your suggestion. We have deleted "to our knowledge" in Line 64.

Q3: Lines 138-141: As mentioned above, I have some concerns regarding the authors' selection of in situ observations. In particular, beyond the underway products already discussed, I was expecting the inclusion of XBT data for identifying frontal structures and validating the presented products. Despite their spatial and temporal limitations, XBT data provide unique insights into the vertical temperature profile of the water column, which is often crucial for accurately locating surface frontal structures, for instance in the Southern Ocean (see Orsi et al., 1995; Rintoul et al., 1997; Budillon and Rintoul, 2003). Since frontal identification is commonly based on temperature values (or gradients) at depths of 150, 300, and 500 m, along depth data could offer valuable complementary information beyond surface-only indicators such as sea surface temperature (SST) to validate the presented product.

The dataset involved in this study, that is the NCEI Surface Underway Marine Database (NCEI-SUMD), contains in situ measurements of SST and SSS, primarily collected through thermosalinographs. It also includes meteorological data from ship-mounted weather packages, microplastic data, and data from unmanned surface vehicles such as Saildrones and Wave Gliders. However, it does not include data from XBT (eXpendable BathyThermograph) probes which provide accurate information about the first hundreds meters of the water column. To access XBT data, for example, the World Ocean Database (WOD) managed by NCEI can be consulted, as it contains temperature and salinity profiles collected from various platforms, including XBTs.

A: Thank you very much for your insightful suggestion. We fully agree that subsurface observations from XBT provide valuable information beyond surface-only indicators for ocean front studies, as demonstrated in the Antarctic front research you mentioned. However, our study primarily relies on satellite observations to identify fronts, meaning that our global front dataset excludes subsurface fronts that lack a clear surface signal. Therefore, incorporating deep-depth fronts observed from XBT would introduce inconsistencies in validation and comparison with our dataset. Moreover, ocean fronts are typically characterized by significant vertical slopes, meaning that the positions of the surface and deep-depth sections of the same front may not align perfectly (Zhang et al., 2024). This positional mismatch could further complicate validation and comparisons. Instead, we believe that integrating the XBT dataset with our dataset presents a valuable opportunity to study the differences between surface and subsurface fronts. However, this topic extends beyond the scope of the present work and warrants further investigation in future research.

Additionally, our validation and comparison require automatic front detection from XBT profile data, yet no established algorithm currently exists for this purpose.

Developing a specialized method for XBT-based front detection would be necessary for future studies. However, previous studies have proposed automatic detection methods for continuous underway observation data (Ullman and Cornillon, 2000; Chang and Cornillon, 2015), which can be effectively applied to our global validation and comparison with satellite-based front detection based on the NCEI-SUMD data. Another key consideration is the spatial resolution difference between datasets. The XBT dataset has a resolution ranging from 25 km to 150 km, whereas our global front dataset, derived from SST, has a finer resolution of 5 km. This discrepancy may limit the ability of XBT data to capture mesoscale fronts characterized by rapid spatiotemporal variations. However, XBT data are well suited for detecting large-scale, relatively stable fronts, such as the Subantarctic Front, Polar Front, and Antarctic Circumpolar Current Front, as noted in the study you referenced. By contrast, the NCEI-SUMD dataset provides near-surface SST observations with spatial resolutions of <1 km and temporal resolutions of <1 hour, offering significant advantages for validating and comparing our mesoscale front dataset.

Given the reasons outlined above, we did not conduct validation and comparison using XBT data. However, we have incorporated discussions in our manuscript to highlight the importance of subsurface fronts and the limitation of our dataset in detecting only surface fronts (Line 399-402), as follows: It should be noted that our front dataset is derived from satellite-observed SST, meaning it excludes some subsurface fronts that lack a clear surface signal. Deep-depth observational profiles from eXpendable BathyThermographs (XBTs) and Argo offer potential for further investigating the differences between our front dataset and subsurface fronts (Rintoul et al., 1997; Budillon and Rintoul, 2003).

- Chang, Y., & Cornillon, P. (2015). A comparison of satellite-derived sea surface temperature fronts using two edge detection algorithms. Deep Sea Research Part II: Topical Studies in Oceanography, 119, 40-47.
- Ullman, D. S., & Cornillon, P. C. (2000). Evaluation of front detection methods for satellite-derived SST data using in situ observations. Journal of Atmospheric and Oceanic Technology, 17(12), 1667-1675.
- Zhang, L., Xu, W., & Li, M. (2024). Frontal slope: A new measure of ocean fronts. Journal of Sea Research, 199, 102493.

A: Thank you for your suggestion, we have revised the sentence (Line 169-170), as follows: The histogram analysis is designed to detect the presence of two distinct

Q4: Lines 168-169: Please, rephrase or clarify.

SST populations using a threshold of >0.75 for the ratio of between-cluster variance to total-cluster variance.

Q5: Section 4.2: I cannot understand if the cross-dataset comparison involves only independent products. Can you clarify on this, please?

A: Thank you for your comment. In this study, we calculated statistical indicators of fronts detected from five independent SST products using underway data to perform a cross-dataset comparison. These SST products include ESA CCI SST, MODIS SST, REMSS SST, GLORYS SST, and HadGEM3 SST. To clarify this, we have added the following sentence (Line 286-288): Aside from our global front dataset derived from the ESA SST product, the hit rate and precision of fronts detected from four other SST products were also individually calculated using underway data to facilitate a comparison of the front detection performance across the five independent SST products.

Q6: Lines 367-368: Please, rephrase.

A: Thank you for your suggestion, we have revised the sentence as follows (Line 365): This study is the first to validate CCAIM on a global scale using worldwide underway observations.

Q7: Figure 3: Please, improve this figure. I cannot read properly the labels and (especially) the colorbar

A: Thank you for your comment. We have increased the label size and adjusted the colorbar position to enhance clarity (Line 205).

Q8: Figure 4: Please, improve colorbar readability

A: Thank you for your comment. We have adjusted the colorbar position to improve clarity (Line 225).

Q9: Figure 5: "the months in the Southern Hemisphere were converted to the Northern Hemisphere by adding 6 months". Is this necessary?

A: Thank you for your comment. We believe this adjustment is necessary. Previous studies have suggested that uniform warming due to strong shortwave radiation and intense stratification during summer may weaken surface frontal intensity, reducing the effectiveness of satellite-based detection algorithms (Yao et al., 2012). To better assess front detection performance across seasons, we adjusted the months

in the Southern Hemisphere by adding 6 months to align them with their Northern Hemisphere counterparts.

Yao, J., Belkin, I., Chen, J., & Wang, D. (2012). Thermal fronts of the southern South China Sea from satellite and in situ data. International Journal of Remote Sensing, 33(23), 7458-7468. https://doi.org/10.1080/01431161.2012.685985.

**Reply to Referee #2**

This paper introduces a valuable global ocean front dataset, employing a refined Cayula-Cornillon algorithm (CCAIM), to provide a daily global high-resolution dataset spanning from 1982 to 2023. The manuscript is well-structured, clearly written, and addresses an important gap in oceanographic research. It also offers rigorous validation against in-situ data and a comprehensive comparison with other datasets. The open access to data and code is commendable and will greatly benefit the scientific community.

Overall, I recommend the manuscript for publication after minor revisions, which are provided below.

A: Thank you for your time and effort in reviewing our manuscript. We have carefully addressed all your concerns and incorporated all your suggestions in the revision.

Q1: There are frequent repetitions of similar concepts, particularly in the Introduction and Discussion sections. I suggest carefully revising these sections to minimize repetition, which would significantly improve readability.

A: Thank you for your suggestion. We have reduced redundancy by removing similar sentences from the Introduction and Discussion sections to enhance readability.

Q2: The manuscript uses the term "multi-satellite merged product" when referring to gap-free (Level 4) SST products. However, in satellite data terminology, "merged multi-sensor" typically refers to Level 3S (super-collated) products. I suggest using either "multi-satellite blended product," "gap-free," "L4 product," or simply "analysis."

A: Thank you so much for your comment. We have replaced all instances of "multisatellite merged" with "multi-satellite blended" in the revised manuscript.

Q3: Lines 24–25: The sentence "...all of which provide independent validation..." is unclear. Please rephrase to improve clarity.

A: Thank you for your comment. We have rewritten it as follows (Line 24-26): Meanwhile, in-situ observations show a strong spatial resemblance to global frontal frequency, providing independent validation of the satellite-based global frontal occurrence map.

Q4: Line 37: Replace "...affect global climate change..." with "...contribute to climate change...".

A: Thank you for your suggestion. We have done it in the revision (Line 38).

Q5: Line 42: The sentence fragment "...ocean fronts oceanographers, climatologists..." seems incomplete or missing an adverb or preposition. Please revise accordingly.

A: Thank you for your careful review. We have incorporated "by" into the sentence (Line 43).

Q6: Line 47: Clarify the term "new technology." Are you referring to the algorithm? If so, consider replacing "technology" with a more specific term, such as "methodology" or "approach."

A: Thank you for your suggestion. We have replaced "technology" with "methodology" in the revision (Line 48).

Q7: Line 58: Replace "accessibility" with "availability."

A: Thank you for your suggestion. We have replaced "accessibility" with "availability" in the revision (Line 59).

Q8: Line 106: Clarify "different types." Types of what?

A: Thank you for your suggestion. We have revised the sentence for clarity as follows (Line 106): Each dataset offers its own advantages and represents a distinct data source.

Q9: Line 130: The phrase "serving as a comparison..." is redundant, as this point has already been mentioned. Consider removing or rephrasing.

A: Thank you for your suggestion. We have revised it (Line 131) as "serving as the output of observation-assimilated models."

Q10: Figure 2 caption: There is a typographical error: "while and d) and e)" should be corrected to remove the extra "and."

A: Thank you for your careful review. We have removed the extra "and" (Line 180).

Q11: Lines 189–190: The sentence starting with "Identifying frontal zones..." is repetitive and not strictly relevant in this section. I recommend removing it.

A: Thank you for your suggestion. We have deleted this sentence in revision.

Q12: Figure 5: The grey shaded area is not described in the caption. Please clarify its meaning explicitly.

A: Thank you for your suggestion. We have included the description to clarify it as follows (Line 277-278): with the grey shaded area representing the number of underway-based frontal pixels.

Q13: Figure 7: I recommend including the ESA CCI hit rate and precision curves here as well, even though they appear in Figure 5, for immediate comparison.

A: Thank you for your valuable suggestion. We have revised Figure 7 based on your suggestions (Line 310).

Q14: Figures 8a, 9a/c, and 10a/c: Figure 8a employs a different map projection from Figures 9 and 10. Please use a consistent map projection to facilitate visual comparisons.

A: Thank you for your suggestion. We have revised Figure 8 based on your suggestions (Line 333).

Q15: Lines 343–344: The sentence starting with "Although ocean fronts..." is unclear. Please rephrase to improve readability.

A: Thank you for your suggestion. We have revised it (Line 346) as: Oceanographic features like fronts cannot be directly observed and are implicitly present in both observed and simulated marine environmental data.

Q16: Line 491: Replace "expected to be widely used" with e.g. "can provide a valuable tool for..."

A: Thank you for your suggestion. We have conducted this modification (Line 492).